# Discovering and Targeting Dynamic Drugging Pockets of Oncogenic Proteins: The Role of Magnesium in Conformational Changes of the G12D Mutated Kirsten Rat Sarcoma-Guanosine Diphosphate Complex

**DOI:** 10.3390/ijms232213865

**Published:** 2022-11-10

**Authors:** Zheyao Hu, Jordi Marti

**Affiliations:** Department of Physics, Polytechnic University of Catalonia-Barcelona Tech, B5-209 Northern Campus UPC, 08034 Barcelona, Catalonia, Spain

**Keywords:** benzothiadiazine derivatives, drug design, molecular dynamics, KRAS oncogenes

## Abstract

KRAS-G12D mutations are the one of most frequent oncogenic drivers in human cancers. Unfortunately, no therapeutic agent directly targeting KRAS-G12D has been clinically approved yet, with such mutated species remaining undrugged. Notably, cofactor Mg${}^{2+}$ is closely related to the function of small GTPases, but no investigation has been conducted yet on Mg${}^{2+}$ when associated with KRAS. Herein, through microsecond scale molecular dynamics simulations, we found that Mg${}^{2+}$ plays a crucial role in the conformational changes of the KRAS-GDP complex. We located two brand new druggable dynamic pockets exclusive to KRAS-G12D. Using the structural characteristics of these two dynamic pockets, we designed in silico the inhibitor DBD15-21-22, which can specifically and tightly target the KRAS-G12D-GDP-Mg${}^{2+}$ ternary complex. Overall, we provide two brand new druggable pockets located on KRAS-G12D and suitable strategies for its inhibition.

## 1. Introduction

KRAS proteins play an important role in various cellular processes, such as cell proliferation, differentiation, and survival [1,2]. As a critical hub in cell signaling networks, KRAS functions as a binary molecular switch [3]: it can be activated by upstream receptors such as EGFR [4,5,6] and can regulate the downstream MAPK [7,8], PI3K [9,10] and RALGDS–RAL [11,12] pathways. KRAS mutations impair both the intrinsic and GAP stimulated GTP hydrolysis activity [13], leading to the hyperactivation of KRAS signaling and ultimately cancer. Plenty of studies have been devoted to the study of the different relevant mutations (G12, G13, Q61 among others) and to the interplay between GDP and GTP [14,15,16,17,18]. After near 40 years of efforts, a major breakthrough was made for KRAS-G12C inhibition [19]. The covalent binding of designed inhibitors to CYS12 induces a targetable allosteric switch-II pocket, eventually blocking KRAS-G12C [20,21,22]. Later on, different new covalent inhibitors targeting KRAS-G12C, such as AMG510 [23,24] and MRTX849 [25], were reported. In 2021, *Sotorasib* was finally approved by the U.S. Food and Drug Administration for the clinical treatment of KRAS-G12C mutant cancers [26,27].

In particular, several strategies have been proposed for targeting KRAS-G12D, such as indole-based inhibitors (BI-2852) of the Switch-I/II pockets [28,29], the piperazine-based compound TH-Z835 for ASP12 [30], the KRAS allosteric ligand KAL-21404358 targeting the site PRO110 [31], a multivalent small molecule named 3144 for pan-RAS inhibition [32] and a three cyclic peptide for GTP-bound KRAS-G12D [33]. It has only been in recent years that more works about G12D, G12R, G12S and G12V were reported and some specific inhibitors were proposed [34,35,36,37,38,39]. Despite these progresses, targeting the most prevalent and oncogenic KRAS-G12D mutation has remained elusive; no therapeutic agent has been clinically approved yet. Therefore, how to interrupt mutated KRAS-G12D oncogene remains a real challenge both in clinical and scientific research.

Clinical data have implicated not only that mutated RAS isoforms vary by tissue and cancer types, but also within the most frequently KRAS-mutations, the mutated amino acid residues of KRAS are also different for each cancer. For instance, among KRAS mutation predominant cancers, KRAS-G12C mutations mainly occur in lung cancer (45%), whereas KRAS-G12D mutations are related to more than 50% of pancreatic ductal adenocarcinoma cases [40,41,42]. The KRAS mutations display high tissue-specific abilities to drive tumorigenesis, strongly suggesting that the mutation of only one KRAS residue can lead to significant mechanistic changes in KRAS behavior. This usually leads to consider that there are only minor differences at the molecular scale between different KRAS mutants, such as one-site GLY12 substitutions. Therefore, the detailed information on atomic interactions and local structures at the all-atom level of KRAS can be of crucial importance for oncogenic KRAS research [43]. Until now, several studies have been published on the effects of G12D mutation on the structure and conformation changes of KRAS [39,44,45,46,47,48]. However, previous studies mainly focus on oncogenic proteins themselves and are less associated with the KRAS surroundings, although cancer is closely related to oncoproteins and the surrounding environment [49,50]. So, more detailed conformations and local structures of KRAS still remain to be revealed.

As one of the important RAS cofactors, the ion Mg${}^{2+}$ is coordinated in an octahedral arrangement with high affinity on RAS proteins [51,52,53]. Mg${}^{2+}$ has been established essential for both guanine nucleotide binding and GTP-hydrolysis of RAS proteins. For instance, in HRAS, the difference in affinity between HRAS and guanine nucleotide in the presence or absence of Mg${}^{2+}$ is ∼500-fold [54,55,56]. However, HRAS is rarely mutated in human cancers with only ∼10% rate found in bladder and cervical cancers [40,41,42]. The mechanism of Mg${}^{2+}$ interaction with the most prevalent oncogenic KRAS species has not been investigated yet. In this paper, using microsecond scale molecular dynamics (MD) simulations at an all-atom level, we reveal that cofactor Mg${}^{2+}$ plays a crucial role in the conformational changes of KRAS. The mutation of GLY12 site in KRAS-G12D triggers a distinct shift in the interaction patterns between Mg${}^{2+}$ and KRAS, finally generating two druggable exclusive dynamic pockets on the KRAS-G12D-GDP-Mg${}^{2+}$ ternary complex surface. In this paper, different from other works in the literature [46], we do not consider the association with the GTP species (which makes KRAS active) since it is related to the signaling of the oncogene, which can occur when attached to the cell membrane surface. Conversely, we consider the KRAS-GDP association (i.e., in the inactive state), which does not require the hugely computationally expensive incorporation of the cell membrane in an explicit way.

With full use of the structural characteristics of these two dynamic pockets we designed in silico the specific inhibitor DBD15-21-22, a derivative of benzothiadiazine [57,58] (DBD), which can specifically and tightly target the KRAS-G12D oncogene, stabilizing the inactive state of KRAS-G12D-GDP. The reliability of this finding was validated by subsequent molecular dynamics simulations of KRAS-G12D together with DBD15-21-22, where, by analyzing the differences between the three-dimensional structure of DBD15-21-22 and guanine nucleotide and combining them with the molecular dynamics simulation results of DBD15-21-22 and wild-type KRAS, we can propose that DBD15-21-22 will be harmless for wild-type KRAS.

## 2. Results and Discussion

Firstly, we investigated the conformational changes of the three different isoforms of KRAS in aqueous ionic solution. The only difference between these three KRAS is located at codons 12, with the sequences and initial structures of the three isoforms shown in Figure A1 (Appendix A). Atomic detailed sketches of GDP and the main residues described in this part are reported in Figure A2. All meaningful atom–atom distances as a function of time and bond lifetimes are reported in Appendix A.

### 2.1. The Fluctuations and Stability of KRAS Protein Conformations with Different Mutation Isoforms

Root mean square deviations and root mean square fluctuations of KRAS-WT and the two mutant isoforms were firstly analyzed (Figure A3A,B). Both properties are defined below. RMSD results show the fluctuations and stability of the conformations of the three species. An overall view of the evolution of conformational changes is shown in Figure A3C–E. We found that for KRAS-WT, there was a distinct conformational fluctuation around 1.9 μs during the simulation. In a similar fashion, for KRAS-G12C, a large conformation change was observed after 1 μs. KRAS-G12D exhibited a distinct conformational stability different from that of KRAS-WT and KRAS-G12C, with overall stability and no significant changes. From the perspective of residues, RMSF reveals its flexibility during the full simulation. Switch-I (SW-I) and Switch-II (SW-II) are the regions of the protein which were recently identified as potential drugging sites [42,48,59], and they show high conformational flexibility compared to other structures of KRAS. Noticeably, we observed that the flexibility of residues in the SW-I domain of KRAS-G12D is around 2-fold smaller than that of KRAS-WT and KRAS-G12C. Combining the RMSD and RMSF results, the KRAS-G12D demonstrated significant stability along full statistics 5 μs MD simulations (2.5 μs per trajectory). This suggests that (1) the different isoforms of KRAS are not sharing the same mechanisms in conformational changes; and (2) it corroborates that to a large extent, the conformational change of the KRAS protein is mainly embodied by SW-I and SW-II. These findings are in overall good qualitative agreement with previous computational studies [60].

### 2.2. Mutation of GLY12 Enhances the Interaction between P-Loop and SW-II Domain

The GLY12 mutation of KRAS protein also shows significant effects on the interaction of P-loop with Switch-II (Figure A4 and Figure A5). GLY12 has weak hydrogen-bonding (HB) interactions with GLY60 and GLN61, but such HB interactions are enhanced when the GLY12 mutated to CYS or ASP. Compared with the GLY12 of KRAS-WT, residue CYS12 contains a sulfhydryl group such that the HB interaction between CYS12 (H1) and GLN61 (O1) is mildly enhanced (see Figure A4B). When GLY12 mutates to ASP, it contains two active HB electron donor sites (carboxyl group). This reverses the surface electrical properties of residue 12, with ASP12 having a longer side chain. The interaction between position 12 and residues GLY60 and GLN61 is significantly enhanced so that they can form three simultaneous strong HB with GLY60 and GLN61 (Figure A5). In addition, the mutation of GLY12 also affects the interaction of VAL8 (near the P-loop) with THR58 (located on SW-II) (Figure A6). In the KRAS-WT case, the $HB_{VAL8(O3)-THR58(H2)}$ and $HB_{VAL8(O3)-THR58(H3)}$ are alternatively generated, but the lifetime of $HB_{VAL8(O3)-THR58(H2)}$ is slightly longer than that of $HB_{VAL8(O3)-THR58(H3)}$. In the KRAS-G12C case, the lifetime of $HB_{VAL8(O3)-THR58(H2)}$ is increased so that such a pair remains present for the entire simulation period, while $HB_{VAL8(O3)-THR58(H3)}$ interactions are mostly eliminated. Furthermore, in the case of KRAS-G12D, the lifetimes of both $HB_{VAL8(O3)-THR58(H2)}$ and $HB_{VAL8(O3)-THR58(H3)}$ are significantly enhanced, with these two HB being present throughout the whole simulation time. Notably, $HB_{VAL8(O3)-THR58(H3)}$ was more stable than $HB_{VAL8(O3)-THR58(H2)}$ in the KRAS-G12D case, which is a clear difference compared to KRAS-WT and KRAS-G12C.

### 2.3. Strong Coordination Interactions between Mg${}^{2+}$ and SER17, ASP57, GDP in KRAS-G12D

Noncovalent interactions, such as hydrogen bonds and coordination bonds (CB), are important for proteins to maintain their tertiary structure as well as for protein–cofactor interactions. We analyzed the structure of the three different KRAS isoforms in the aqueous ionic solution by time-dependent atomic site–site distances between selected atomic sites to find out and estimate interactions between Mg${}^{2+}$ and KRAS (Figure A7). The full set of averaged values and lifetimes of HB/CB are also reported in Table A2 and Table A3 of Appendix A. We selected in Figure 1 representative snapshots from the total computed length of 5 μs, where the pattern and mechanism of the formation and breaking of CB involving Mg${}^{2+}$ are clearly seen. Interestingly, cofactor Mg${}^{2+}$ exhibits a unique role in the conformational change of KRAS-G12D, while having little influence on KRAS-WT and KRAS-G12C. As shown in Figure 1, for the KRAS-G12D case, Mg${}^{2+}$ can form CB with SER17 (oxygen labeled ‘O2’, see Figure A2), ASP57 (O1, O2) of KRAS-G12D and oxygens O1B and O2B of GDP (state 1). During the first 500 ns, $CB_{Mg^{2+}-SER17\left(O2\right)}$ is in a dynamic fluctuation and can be formed and broken (state 2). In the next 1.5 μs, bonds $CB_{Mg^{2+}-GDP\left(O2B\right)}$ and $CB_{Mg^{2+}-ASP57\left(O2\right)}$ are broken. Finally, Mg${}^{2+}$ forms strong CB with GDP (O1B), ASP57 (O1) and SER17 (O2) (state 3). In KRAS-WT and KRAS-G12C cases, the interaction patterns of Mg${}^{2+}$ with KRAS and GDP are similar to each other, but they are very different from the KRAS-G12D case. In particular, Mg${}^{2+}$ forms CB with SER17 (O2) and O2A, O1B, and O2B atoms of GDP, while the lifetimes of Mg${}^{2+}$-SER17 (O2) and Mg${}^{2+}$-GDP (O1B) are rather short. Once the $CB_{Mg^{2+}-SER17\left(O2\right)}$ bond was broken, Mg${}^{2+}$ was subsequently released from the binding site of KRAS-SER17.

### 2.4. GDP Plays an Important Regulatory Role in the Conformational Change of SW-I

In addition to the aforementioned CB/HB interactions in KRAS-GDP-Mg${}^{2+}$ complex, the HB interactions were also investigated between SW-I and GDP (Figure A8). We found that the HB interaction of SW-I (ASP30) with GDP is important for the transition of the SW-I conformation. The H2 and H3 atoms of GDP are very sensitive: when they form hydrogen bonds with the oxygen atoms of ASP30, these HBs between ASP30 and GDP can limit the conformational change of SW-I within a small range. From the time evolution of selected atom–atom distances d(t) in Figure A8, we can see that the strength of the HB interaction between ASP30 and GDP are KRAS-G12D > KRAS-WT > KRAS-G12C. This matches the RMSD variation of each KRAS in Figure A3A. Correspondingly, KRAS-G12D exhibits extreme conformational stability during intervals of at least 2.5 μs of the simulations. We also observed SW-I of KRAS-WT opening largely, while the HB interaction between ASP30 and GDP breaks around 1.9 μs. Finally, since ASP30 has the weakest HB interaction with GDP in the KRAS-G12C case, the latter exhibits large conformational changes around 1 μs.

### 2.5. The Dominant Conformations of Wild-Type and Mutated KRAS

Gibbs free energy profiles are of high significance to characterize the dominant conformations of KRAS during simulation. In the present work, such analytical tools will allow us to directly track the effects of GLY12 mutations on the dominant conformations of the KRAS. Since the main movement of the protein is concentrated in SW-I and SW-II and our main research object is KRAS-GDP-Mg${}^{2+}$ ternary complex, we chose to compute Gibbs free energy landscapes by using two specific variables, including RMSD and radius of gyration, see below. The method employed to obtain the free energy profiles is the so-called “principal component analysis” [61,62], where the components RMSD and $R_{g}$ work as reaction coordinates (Figure 2). In the KRAS-WT case, we detected two free energy basins (I, II). Basin II is the one with lowest free energy (set to 0 kJ/mol), and it was chosen as the reference so that basin I shows a barrier of 4.6 kJ/mol, when transitions from basins I to II are considered. These two Gibbs free energy basins represent the two main stable states of KRAS-WT during MD simulations. We can obtain from the MD trajectories that in state I, SW-I is slightly open, while in the dominant basin II, SW-I opens largely and SW-II tends to be closer to the P-loop (region containing the mutated codon 12 and depicted in orange color). Several Gibbs free energy basins were also detected in the KRAS-G12C. In this case, the representative conformation of basin I is similar to the corresponding one in KRAS-WT, while basin II is divided into several small energy basins, with IIa and IIb as examples. The dominant conformation is now located on basin IIb (0 kJ/mol), with basin I at 2.7 kJ/mol and basin IIa at 1.0 kJ/mol. These basins are separated by low free energy barriers and they are moderately easy to access by the system. The differences between basins IIa and IIb are mainly on SW-I, with the SW-I conformation of IIb being more open than that of IIa. The comparison between KRAS-WT, KRAS-G12C and KRAS-G12D shows a clear difference: a single Gibbs free energy basin is found for KRAS-G12D, which indicates that such species have only one dominant stable conformation. As it can be seen from the representative crystal structure of this particular free energy basin I (Figure 2C), this dominant conformation of KRAS-G12D corresponds to SW-I being slightly open, while the distance between SW-II and P-loop is relatively tight. This is because the strong CB interaction of Mg${}^{2+}$ with GDP (O1B), SER17 (O2) and ASP57 (O1) can stabilize the distance between SW-II, GDP and the α-helix of KRAS, where SER17 is located. We also observed that the ASP12 shows strong interaction with GLY60 and GLN61, both located at SW-II. At the same time, hydrogen atoms H2 and H3 of GDP can form stable HB interactions with ASP30, helping to stabilize the distance between SW-I and GDP. To corroborate the full convergence of the free energy landscapes reported in Figure 2, we report in Appendix A (Figure A9) the contribution of the two independent 2.5 μs trajectories that we employed in this work (A,B), compared with their average (C), as shown in Figure 2C.

### 2.6. The Two Unique Druggable Dynamic Pockets on KRAS-G12D

From the previous analysis, KRAS-G12D exhibited a stable conformation that was significantly different from those of KRAS-WT and KRAS-G12C, and Mg${}^{2+}$ exhibited a unique binding mode in the KRAS-G12D surface. Mg${}^{2+}$ can form strong CB interactions with GDP (O1B), SER17 (O2) and ASP57 (O1). Well-established experimental results [51,52] showed that the octahedral structure is the most stable structure for Mg${}^{2+}$ first shell coordination, which is fully consistent with our simulation results. When we take into account the interaction of KRAS-GDP-Mg${}^{2+}$ complexes in aqueous Mg-Cl solution, we find the dynamic water pocket I in the Mg${}^{2+}$ binding area (left side of Figure 3). We call it “dynamic” in the following sense: in the octahedral structure of Mg${}^{2+}$, the fluctuation frequencies of the coordinated water molecules (in positions H${}_{2}$O-1, H${}_{2}$O-2 and H${}_{2}$O-3) are lower than those of free water molecules in solution, i.e., the former remain in their positions for much longer periods of time than the latter (nanoseconds compared to the picosecond time scale for bulk water HB dynamics [63]). At the same time, the coordinated water molecules can be exchanged with other water molecules in the solution (Figure A10). Further analysis of the trajectory uncovered another dynamic water pocket II, which is located between the phosphate group of GDP and the ASP12 (right side of Figure 3). This dynamic pocket can accommodate only one H${}_{2}$O molecule, through HB interactions of H${}_{2}$O with GDP (O2B) and ASP12 (O1/O2) (Figure A11). Further, the water molecule in this pocket is in a dynamic fast exchange with other water molecules in the solution. For the sake of corroboration of the newly observed pockets I and II, we included in Appendix A (see Figure A12) a comparison of the location of the pocket reported by Kessler et al. [28] with the pockets reported in the present work.

### 2.7. Structure-Based Drug Design and Targeting the Dynamic Water Pockets on KRAS-G12D

In order to target the newly revealed binding pockets, we designed a drug in silico able to block GDP at its main cavity in a permanent way. There are several identified ways to target KRAS, the following being the most relevant [64]: inhibition of RAS expression, interference with RAS post-translational modifications, inhibition of RAS function or targeting specific downstream effectors. Furthermore, our strategy is different from that of the designers of recent drug *Sotorasib* [65]. We observed that in recent years, selective binding ligands for Mg${}^{2+}$ have gradually become known [66]. In dynamic water pocket I, H${}_{2}$O-2 and H${}_{2}$O-3 molecules can exchange with other water molecules in the aqueous solution. The two oxygen atoms of H${}_{2}$O-2 and H${}_{2}$O-3 are on the same horizontal plane, and the distance between the two oxygen atoms is comparable to the size of the bidentate ligand. This feature makes it possible to design the selective inhibitors that can replace these two water molecules and target this dynamic water pocket I. Herein, we designed a series of inhibitors for these two dynamic water pockets using a benzothiadiazine, specifically 3,4-dihydro-1,2,4-benzothiadiazine-1,1-dioxide (C_7_H_8_N_2_O_2_S, DBD), as a template. The most suitable species obtained is designed as DBD15-21-22 (see Figure 4A) since it is a DBD derivative, similar to some of those reported in Ref. [67].

### 2.8. DBD15-21-22 Can Target KRAS-G12D and Bind Dynamic Water Pockets I and II

To further study the structure of the DBD15-21-22 interaction with KRAS-G12D, we performed microsecond time scale MD simulations of DBD15-21-22 together with KRAS-G12D. As we can see from the MD results, the DBD15-21-22 is tightly bound to its binding pocket (Figure 4B). The N4 and N5 of DBD15-21-22 can replace H${}_{2}$O-2 and H${}_{2}$O-3 in the octahedral structure of Mg${}^{2+}$ and then specifically bind to the dynamic water pocket I. Benefiting from the location and size of the amino group ($-NH_{2}$) near the N4 and N5, the $-NH_{2}$ group can perfectly replace the H${}_{2}$O-4 in the dynamic water pocket II. These two concerted actions enable DBD15-21-22 to bind tightly and specifically to the two dynamic water pockets on the KRAS-G12D-GDP-Mg${}^{2+}$ complex. In addition, other active sites in DBD15-21-22 can also form stable HB interactions with other residues of KRAS-G12D. In particular (see Figure 4A), the H1 atom of DBD15-21-22 can form HB with ASP57 (O2) and the H11 atom on the phenolic hydroxyl (−OH) group can form HB with multiple residues on SW-I to lock the conformational change of SW-I. In order to estimate the detailed HB and CB interactions between DBD15-21-22 and its unique druggable pocket on KRAS-G12D, we display the time evolution of selected atom–atom distances d(t) in Figure A12, Figure A13, Figure A14, Figure A15, Figure A16, Figure A17 and Figure A18.

### 2.9. DBD15-21-22 Is Harmless to KRAS-WT

As we can see in Figure 5A, in the structure of DBD15-21-22 there exists a guanylate moiety but the binding patterns are quite different from GDP/GTP. When GDP/GTP binds to the GDP/GTP binding pocket (Figure 5E), the O3 of guanine goes deep inside the pocket. DBD15-21-22 also has a guanine group, which is similar to GDP/GTP. The difference is that in DBD15-21-22 (Figure 5A), the DBD part is connected to the N3 atom of guanine; in contrast, the β-D-ribofuranosyl in the GDP structure is connected to the N4 atom of guanine (Figure 5C). The drug we designed takes advantage of this subtle difference, and prevents the guanine structure in the DBD15-21-22 from binding to the GDP/GTP pocket through the steric hindrance of the DBD part in the DBD15-21-22. Moreover, we investigated the three-dimensional structure of DBD15-21-22 and GDP in an aqueous solution (Figure 5D,F). DBD15-21-22 can form intramolecular hydrogen bonds (Figure 5D), further preventing DBD15-21-22 from binding to the binding site of GDP/GTP on KRAS, and thereby minimizing the toxicity to KRAS-WT.

To further verify the low toxicity of DBD15-21-22 to KRAS-WT, we also performed MD simulations of DBD15-21-22 together with GDP free KRAS-WT. From the simulation results, we can see that DBD15-21-22 was unable to bind to the GDP/GTP binding pocket on KRAS-WT but was dissolved in an aqueous solution most of the time (Figure A19A).This finding is further verified by the radial distribution function (RDF) of the DBD15-21-22 interaction with aqueous solutions (Figure A19B). We can observe that DBD15-21-22 dissolves well in water, with hydrogen H1 and nitrogen N5 of DBD15-21-22 both forming HBs with H${}_{2}$O. However, in the KRAS-G12D-DBD15-21-22 simulation case, the HB interaction of DBD15-21-22 with water was interrupted such that DBD15-21-22 was well bound to the druggable pocket on the surface of KRAS-G12D, as described in Figure 4B).

## 3. Methods and Materials

Our main computational tool has been microsecond scale molecular dynamics. In MD, after the choice of reliable force fields, the corresponding Newton’s equations of motion are integrated numerically [68], allowing us to monitor each individual atom in the system in a wide variety of setups, including liquids at interfaces in solid walls or biological membranes, among others [69,70]. We fixed the number of particles, the pressure and the temperature of the system, while the volume was adjusted accordingly. MD can model hydrogens at the classical [71,72] or quantum levels [73] and, in addition to energetic and structural properties, MD provides access to time-dependent quantities, such as the diffusion coefficients, rotational and vibrational motions [74] or spectral densities [75], enhancing its applicability. In the present work, we conducted MD simulations of three KRAS isoforms with sequences represented in Figure A1. Each system contains one isoform of KRAS-GDP complex fully solvated by 5697 TIP3P water molecules [76] in potassium chloride solution at the human body concentration (0.15 M) and magnesium chloride solution concentration (0.03 M), yielding a system size of 19,900 atoms. All MD inputs were generated using the CHARMM-GUI solution builder [77,78,79], and the CHARMM36m force field [80] was adopted for KRAS-GDP-Mg${}^{2+}$ interactions. The force field used also includes the parameterization of the species GDP (it can be searched as “GDP” in the corresponding CHARMM36m topology file: https://www.charmm-gui.org/?doc=archive&lib=csml (accessed on 10 October 2021)). All bonds involving hydrogens were set to fixed lengths, allowing fluctuations of bond distances and angles for the remaining atoms. The crystal structure of GDP-bound KRAS proteins was downloaded from the RCSB PDB Protein Data Bank [81], file name “4obe”. The three sets of KRAS-GDP complex (wild type, G12C mutate and G12D mutate) were solvated in a water box, all systems were energy minimized for 50,000 steps and well equilibrated (NVT ensemble) for 250 ps before generating the production MD. Two independent production runs were performed within the NPT ensemble for 2.5 μs. All meaningful properties were either averaged from the two runs.The pressure and temperature were set at 1 atm and 310.15 K respectively, in order to simulate the human environment. In all MD simulations, the GROMACS/2021 package was employed [82]. Time steps of 2 fs were used in all production simulations, and the particle mesh Ewald method with a Coulomb radius of 1.2 nm was employed to compute long-ranged electrostatic interactions. The cutoff for Lennard–Jones interactions was set to 1.2 nm. Pressure was controlled by a Parrinello–Rahman piston with damping coefficient of 5 ps${}^{-1}$, whereas temperature was controlled by a Nosé–Hoover thermostat with a damping coefficient of 1 ps${}^{-1}$. Periodic boundary conditions in three directions of space were taken. We employed the “gmx-sham” tool of the GROMACS/2021 package to perform the Gibbs free energy landscape analysis. Other alternatives, such as transition path sampling [83,84,85] or metadynamics [86,87], were not considered here because of their high computational cost for large systems. The designed inhibitor DBD15-21-22 was parameterized using CgenFF [88,89] to obtain the CHARMM-compatible topology and parameter files. Further, we switched to run another four 1 μs of MD simulations to study the interaction of the inhibitor DBD15-21-22 with KRAS-G12D and to verify that DBD-15-21-22 is harmless on the wild-type KRAS. These four new MD simulations have the same setup as the previous ones. Moreover, the software VMD [90] and UCSF Chimera [91] were used for trajectory analysis and visualization.

As key computed quantities in this work, the radius of gyration ($R_{g}$) is defined in Equation (Equation 1):(1)$$R_{g}=\sqrt{\frac{\sum_{i}{∥r_{i}∥}^{2}m_{i}}{\sum_{i}m_{i}}},$$
where $m_{i}$ is the mass of atom *i*, and $r_{i}$ the position of the same atom with respect to the center of mass of the selected group. Root mean square deviations (*RMSD*) are defined by Equation (Equation 2):(2)$$RMSD\left(t\right)\equiv \sqrt{\frac{1}{N}\sum_{i=1}^{N}\delta_{i}^{2}\left(t\right)},$$
where $\delta_{i}$ is the difference in distance between the atom *i* (located at $x_{i}\left(t\right)$) of the catalytic domain and the equivalent location in the crystal structure, and finally root mean square fluctuations (*RMSF*) are defined by Equation (Equation 3):(3)$$RMSF_{i}\equiv \sqrt{\frac{1}{\Delta t}\sum_{t_{j}=1}^{\Delta t}{\left(x_{i}\left(t_{j}\right)-{\tilde{x}}_{i}\right)}^{2}},$$
where ${\tilde{x}}_{i}$ is the time average of $x_{i}$, and Δt is the time interval where the average is taken.

## 4. Conclusions

To summarize the main findings of the work, let us remark that KRAS mutations are widely present in many cancer cases, but there exist usually only molecular-scale differences between different KRAS mutants, such as in the one site GLY12 mutation. This means the direct information on atomic interactions and local structures at the all-atom level of KRAS can be extremely useful for oncogenic KRAS research. Based on powerful computer simulation tools, we focused our analysis on the local structure of KRAS proteins when associated with Mg${}^{2+}$, GDP, and water. After systematic analysis of meaningful equilibrated data, we can reveal the dynamic differences between KRAS-WT and the mutated species KRAS-G12C and KRAS-G12D at the all-atom level. There exist several important intermolecular interactions affecting the behavior of KRAS. The most relevant features are (1) the strong coordination interactions between Mg${}^{2+}$, SER17, ASP57 and GDP in KRAS-G12D; (2) the mutation of GLY12 enhances the interaction between P-loop and SW-II domain, with the enhancement magnitude ordered as KRAS-G12D > KRAS-G12C; and (3) the GLY12 mutation also affects the interaction between GDP and SW-I. Although GLY12 mutation can enhance the interaction of the P-loop with SW-II, in the KRAS-G12C case, the interaction of GDP with SW-I was almost totally eliminated, whereas the G12D mutation greatly enhanced the interaction of GDP with SW-I. In Figure 6, we show the alignment of three representative dominant structures of the GDP-bound KRAS-WT, KRAS-G12C and KRAS-G12D. As shown in Figure 6A, the downward sliding of the α-helix and β-sheet structures of KRAS-WT promoted the large opening of the SW-I domain. When GLY12 was mutated to ASP12, the interaction between P-loop and SW-II was largely enhanced, then the relative position of P-loop and SW-II was shortened and stabilized. This promotes strong coordination interactions between Mg${}^{2+}$, SER17, ASP57 and GDP, which further stabilizes the relative positions of β-Sheet and SW-II in the protein structure. Also benefiting from the hydrogen bond interaction between ASP30 and GDP, the fluctuation of SW-I was limited to a small range, which led to a slightly open stable conformation of SW-I in KRAS-G12D. When GLY12 was mutated to CYS12, although the interaction between P-loop and SW-II was enhanced, it was not enough to stabilize the relative positions of α-Helix, β-Sheet and SW-II in the structure of KRAS, and the conformational changes were similar to those of the KRAS-WT.

After deciphering the laws of conformational changes of different KRAS mutants, two druggable dynamic water pockets on KRAS-G12D-GDP-Mg${}^{2+}$ ternary complex were revealed. We designed a specific inhibitor DBD15-21-22 for these dynamic water pockets, with the aim to lock the KRAS-G12D in the “off” (inactive) state, i.e., when the protein is unable to perform signaling processes and eventually develops cancer. Benefiting from the fact that H1 and O3 atoms of DBD15-21-22 can form intramolecular hydrogen bonds in aqueous solution, we detected no binding effects of DBD15-21-22 toward the GDP/GTP binding pocket of KRAS-WT, whereas our molecular dynamics simulation results show that this KRAS-G12D inhibitor can efficiently bind to the targeting pocket located between SW-I, SW-II and P-loop, thereby efficiently locking KRAS-G12D in the “off” state. Overall, our work provides a new druggable pocket and a reliable protocol for the development of specific inhibitors targeting KRAS-G12D.

## Figures and Tables

**Figure 1 ijms-23-13865-f001:**
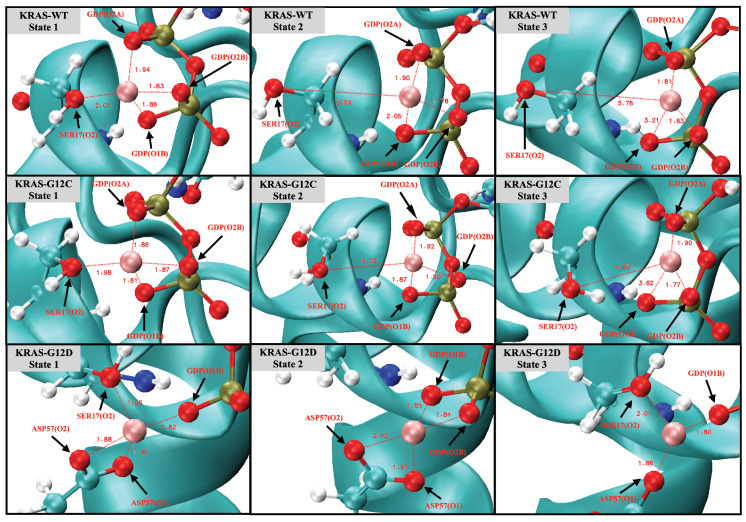
Snapshots of the coordination bonds between Mg${}^{2+}$ and SER17, ASP57 and GDP. Water molecules are hidden for the sake of clarity.

**Figure 2 ijms-23-13865-f002:**
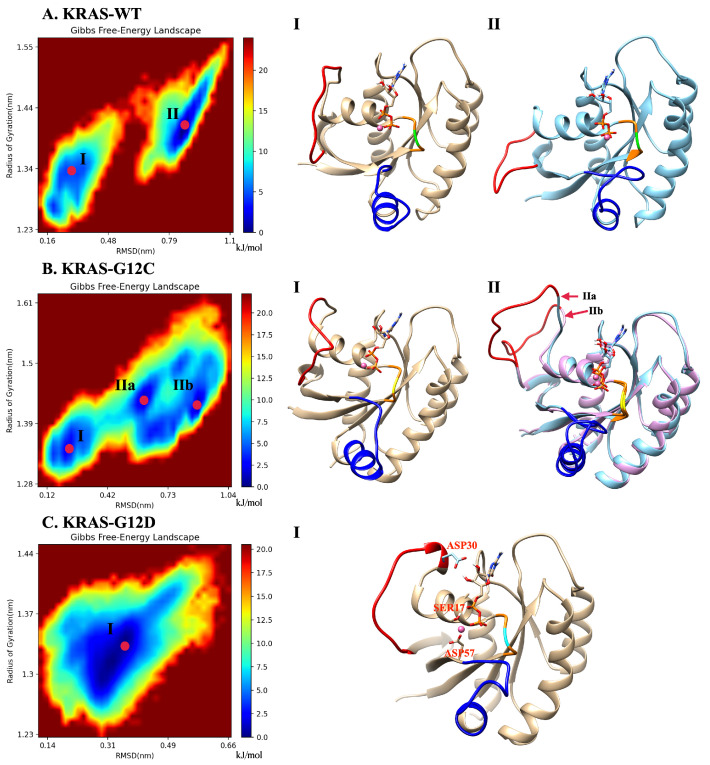
Gibbs free energy landscapes and representative conformations: (**A**) landscapes of the KRAS-WT-GDP-Mg${}^{2+}$ complex and the representative dominant conformations of basins I and II; (**B**) landscapes of KRAS-G12C-GDP-Mg${}^{2+}$ complex and the representative dominant conformations of basins I and II; (**C**) landscapes of the KRAS-G12D-GDP-Mg${}^{2+}$ complex and the single representative dominant conformation (I). Selected groups: backbone atoms of residue 10 to 72; GDP and the Mg${}^{2+}$ binding on KRAS. Red: SW-I; Blue: SW-II; Orange: P-loop; Green: GLY12; Yellow: CYS12; Cyan: ASP12.

**Figure 3 ijms-23-13865-f003:**
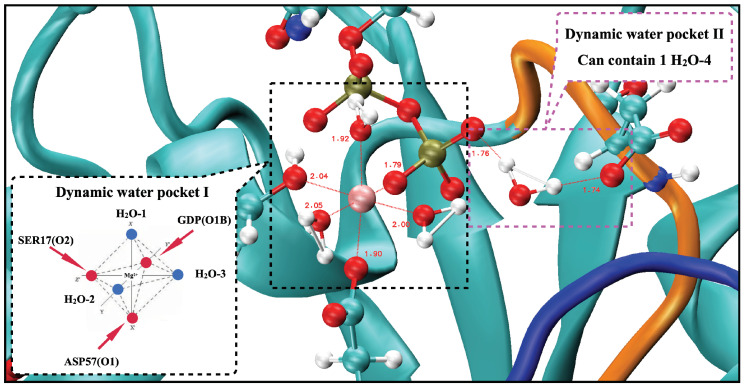
Snapshot of two dynamic pockets on the surface of KRAS-G12D-GDP-Mg${}^{2+}$ ternary complex.

**Figure 4 ijms-23-13865-f004:**
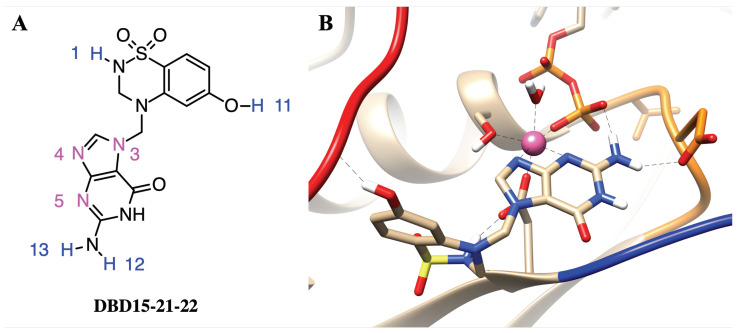
(**A**) Chemical structure of designed inhibitor DBD15-21-22 and (**B**) sketch of the interaction of details DBD15-21-22 with KRAS-G12D. Colors as in Figure 2.

**Figure 5 ijms-23-13865-f005:**
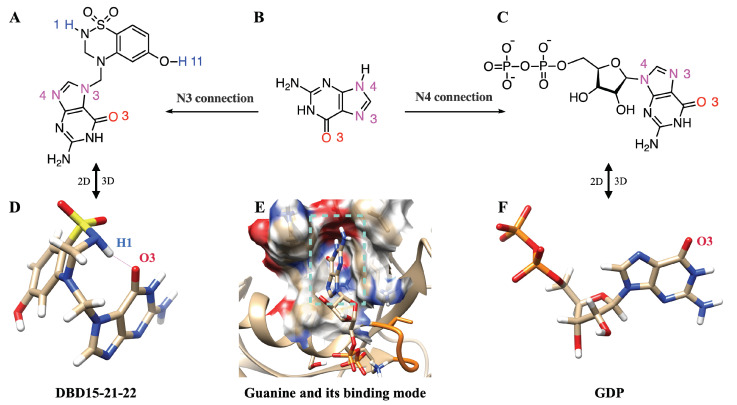
DBD15-21-22 cannot bind to the GDP/GTP binding pocket of KRAS-WT: (**A**) 2D structure of DBD15-21-22; (**B**) 2D structure of guanine; (**C**) 2D structure of GDP; (**D**) 3D structure of DBD15-21-22 in water; (**E**) GDP-binding model of KRAS-WT; (**F**) 3D structure of GDP in water.

**Figure 6 ijms-23-13865-f006:**
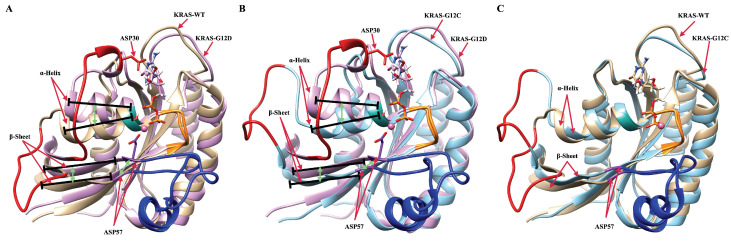
Alignment of three representative dominant structures of the GDP-bound KRAS-WT, KRAS-G12C and KRAS-G12D. (**A**) Alignment of GDP-bound KRAS-WT and GDP-bound KRAS-G12D; (**B**) alignment of GDP-bound KRAS-G12C and GDP-bound KRAS-G12D; (**C**) alignment of GDP-bound KRAS-WT and GDP-bound KRAS-G12C.

## Data Availability

Not applicable.

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
