# Peer review of "Discovering and Targeting Dynamic Drugging Pockets of Oncogenic Proteins: The Role of Magnesium in Conformational Changes of the G12D Mutated Kirsten Rat Sarcoma-Guanosine Diphosphate Complex"

_ijms, 2022, doi:10.3390/ijms232213865_

Round 1
Reviewer 1 Report
This is a very good and relevant paper, addressing a topical problem. It is expertly executed too, which is not surprising considering that it is coming from this group. I only have a few minor points. These concern updates.
It seems like the authors did not follow or update the references. Currently, the statement that no inhibitors have been approved is incorrect. Sotorasib has been approved by the FDA and is already in the clinics for some time. Drugs are also in the pipeline (mostly be Kevan Shokat's group, or others who follow his work) for G12D, G12R, G12S, etc. Also, reversible drugs are in the pipeline as well, and some nucleotide-independent, for the K-Ras inactive (GDP-bound) and K-Ras active, GTP-bound state, too. He discovered and is using two binding sites as well. Please check his publications and update the narrative as needed. Also related to the manuscript here is the NMR and MD observation of the behavior of the inactive K-Ras G12D mutant (PMID: 35752212).
Altogether, an excellent, rigorous work, that once updated will be an important contribution to the field.
Author Response
First of all we warmly thank the referee for his/her positive report. We have prepared this work for quite a long time and we agree we still missed to update a few recent findings (mostly of 2022) in the “Introduction”. Such drawback has been revised and corrected. We have included new refs. : 23, 25-27, 29, 34-39 and 43, related to recent findings on mutations G12D, G12R, G12S, G12V and new technical developments.
Reviewer 2 Report
Article entitled “Discovering and targeting dynamic drugging pockets of oncogenic proteins: the role of magnesium in conformational changes of the G12D mutated Kirsten Rat Sarcoma-guanosine diphosphate complex” found two brand new druggable dynamic pockets exclusive to KRAS-G12D. Using the structural characteristics of these two dynamic pockets and identified inhibitor DBD15-21-22, which can precisely and tightly target KRAS-G12D-GDP-Mg2+ ternary complex. Overall, this article provides two new druggable pockets on KRAS-G12D and suitable strategies for its inhibition. Same research group predicted the conformational variability of oncogenic gtp-bound g12d mutated KRAS-4B proteins at zwitterionic model cell membranes (DOI: 10.1039/D1NR07622A). Overall this paper is well-written and discussed. This article may be considered for publication.
Comments
· Author performed microsecond scale molecular dynamics to investigate the conformational changes of the three different isoforms of 129 KRAS in aqueous ionic solution and found distinct conformational stability different from that of KRAS-WT and KRAS-G12C. Have you checked the HBond network and compared it with the G12C?
· Sotorasib is KRAS Inhibitor in FDA approval stage, Can it be used for G12D?
· Mg2+ role in drug inhibition?
Author Response
We sincerely thank the referee for his/her positive report. On the questions, we believe all of them have been taken into account in the manuscript, so we just would like to clarify such points below:
- On the analysis of the HBond network, we focus our work on the water first coordination shell of magnesium, since these few waters were the relevant ones for this study. We compared wild-type, G12D and G12C mutations. The remaining water molecules in solution were just solvating the system but not playing a relevant role on the GDP blocking process and were not analysed in detail. We simply considered water solvation of the new drug DBD-15-21-22 in Fig. A25.
- About the use of sotorasib for the tumours based on mutation G12D, we believe it is not suitable, since this drug was specifically designed for the mutation G12C which specifically modifies the local structure of the p-loop mutated region and of Switches I and II in a neatly different way as G12D does. For such reason drugs are designed targeting only one mutation. This can be clearly seen from Refs. 14-19 and 34-39.
- The role of magnesium ion in drug inhibition should be described as a key agent, not only for the GDP-GTP exchange that allows the activation of the tumour but also for the particular species that we propose as a new blocking drug. This has been described with details in Sections 3.3, 3.8 and in Section 2 of the Appendix.